# Responses of Humoral and Cellular Immune Mediators in BALB/c Mice to LipX (PE11) as Seed Tuberculosis Vaccine Candidates

**DOI:** 10.3390/genes13111954

**Published:** 2022-10-26

**Authors:** Andriansjah Rukmana, Lulut Azmi Supardi, Fithriyah Sjatha, Mifa Nurfadilah

**Affiliations:** 1Department of Microbiology, Faculty of Medicine, Universitas Indonesia, Jakarta 10320, Indonesia; 2Masters’s Programme in Biomedical Sciences, Faculty of Medicine, Universitas Indonesia, Jakarta 10430, Indonesia

**Keywords:** tuberculosis, LipX (PE11), mice

## Abstract

A member of the *pe*/*ppe* gene family, *lipX* (*pe11*), is capable of directing persistent *Mycobacterium tuberculosis* and avoiding host immune responses. Some studies have indicated that LipX (PE11) can detect humoral antibodies in tuberculosis patients. Hence, information on immune mediators’ responses to this protein is essential to understand its protective efficacy against *M. tuberculosis* infections. This study aimed to examine the response of immune mediators to pCDNA3.1-*lipX* expression in vivo. In the experiment, pCDNA3.1-*lipX* was injected into BALB/c strain male mice aged between 6 and 8 weeks, and they were compared to groups injected with pCDNA3.1 and without injection. The injection was carried out three times intramuscularly every two weeks. Blood was taken retro-orbitally and used for humoral response analysis by Western blotting against LipX-His protein. Simultaneously, the splenocytes were cultured and induced with LipX-His protein for cellular immunity analyses. Our study showed that the recombinant DNA of pCDNA3.1-*lipX* induced a humoral and cellular immune response, especially in IL-4, IL-12, and IFN-γ, which are the primary cellular responses to *M. tuberculosis* infections. However, additional studies, such as a challenge study, are needed to strengthen the argument that this plasmid construction is feasible as a tuberculosis seed vaccine candidate.

## 1. Introduction

Tuberculosis (TB) is an infectious disease that has become a persistent global health problem, especially in developing countries. The World Health Organization (WHO) reports 10.4 million new TB cases annually, with a mortality rate of 1.8 million worldwide [1,2,3]. *M. tuberculosis* infection is the tenth-highest cause of death in the world. In Indonesia, it is estimated that one million new TB cases occur every year [2,4].

The most effective intervention for suppressing TB cases is vaccination, rather than drugs. The anti-TB vaccine currently being used is Bacillus Calmette–Guerin (BCG) [5,6,7]. However, the BCG vaccine is less able to protect adults and immunocompromised or HIV-positive individuals [5]. HIV-positive cases often result in coinfection with TB [4,5,6,8]. The BCG vaccine has varying efficacy against pulmonary tuberculosis infection, especially in developing countries and populations infected by the *M. tuberculosis* Beijing strain [1,6,9]. Thus, a study of new TB vaccine candidates is needed [8,10].

DNA vaccines consist of vectors that contain specific antigen-encoding proteins from specific microbes [8]. DNA vaccines are a safer form of vaccine to give to various people. This kind of vaccine can stimulate the host’s humoral and cellular immune responses to various diseases [11,12,13,14]. The pcDNA3.1 vector is a plasmid commonly used in recombinant protein expression in mammalian cells. This plasmid is also widely used to develop recombinant DNA-based vaccines for infectious diseases and other purposes, such as cancer gene therapy [15,16,17,18]. To construct a DNA vaccine, specific antigens that can induce an immune response are needed [19].

The family of *pe*/*ppe* genes occupies 8–10% of the *M. tuberculosis* DNA genome. This family is predicted to contribute to the level of *M. tuberculosis* virulence [20,21,22,23]. One member of this family, *lipX* (*pe11*), is known to have a significant role in directing the occurrence of infection, and it is considered one of *M. tuberculosis*’s main factors in hindering host immune evasion, especially by macrophage cells [24,25]. LipX is present in some pathogenic types of *Mycobacterium,* like *M. tuberculosis*, *M. bovis*, and CDC1551, a clinical isolate, but is absent in nonpathogenic strains of *M. smegmatis* [25]. In silico analysis has shown that LipX can induce the adaptive host immune response, which enables its other potential function as a biomarker in extrapulmonary TB in children [26].

*M. tuberculosis* infection control depends on the development of an adaptive immune response in the humoral and cellular stages at the site of infection in the host [27]. Immunity to TB infection is predominantly mediated by cellular immune responses and interactions between CD4+ lymphocytes and macrophage cells. These cells interact with each other via a complex network of cytokines. T helper 1 (Th1) cells are considered necessary because they produce cytokines IL-2 and IFN-γ, thus enabling the host to fight *Mycobacterium* infection by activating macrophages. IL-12 is a cytokine that is important in the maturation and development of lymphocytes. In contrast, Th2 secretes IL-4 to regulate lesions. The IL-10 from regulated cells inhibits the activity of CD4+, IL-12, and macrophages and then induces the host to fail against infection [28,29].

In this study, we analyzed the cellular immune response of IFN-γ, IL-4, IL-10, and IL-12 to LipX protein in a BALB/c mice strain. Mice were used to evaluate the efficacy of this new vaccine candidate because of their low cost, feasibility of use in the laboratory, very well-characterized immune system, and abundant available commercial reagents [30]. Following BCG immunization, as a resistant mouse strain, BALB/c showed protective effects induced by the vaccine, which also occur in humans [31]. Mouse models can tell us about the immune mechanisms associated with the effectiveness and application of vaccines in humans. 

## 2. Materials and Methods

### 2.1. Strains

*M. tuberculosis* Beijing and the top 10 *Escherichia coli* strains were used in this study. Both strains are culture stocks of the Microbiology Department, Faculty of Medicine, Universitas Indonesia.

### 2.2. Plasmid and Protein Recombinants

We used recombinant plasmids pET28a-*lipX* (unpublished data) and pCDNA3.1-*lipX*, which were constructed in our previous work [32]. All plasmids are stocks of the Department of Microbiology, Faculty of Medicine, Universitas Indonesia (Appendix B).

### 2.3. LipX-His Protein Isolation

The *E. coli* BL21 strain harboring pET28a-*lipX* was grown in Luria Bertani (LB) medium (merck) and induced by 1 mM of IPTG for 4 h in a shaker incubator at 37 °C at 200 rpm, which was followed by protein isolation using an Ni-NTA affinity column (Qiagen). Isolated LipX-His protein was then confirmed by Western blot using mouse antibody monoclonal anti-His (Thermo Fisher Scientific, Waltham, MA, USA). The recombinant LipX-His protein was stored for further use in the analysis of the humoral response and as an inducer of a cellular immune response from mice immunized with pcDNA3.1-*lipX* recombinant DNA.

### 2.4. Sample Size and Animal Care

This experiment used the BALB/c mouse strain from AniLab Bogor. The mice were male and aged 6–8 weeks. The number of test animals used for each test group was determined using the Federer formula. Federer’s formula is (n − 1) (t − 1) > 15, where the t value is the number of treatment groups and the n value is the number of animals per treatment group. Based on this formula, three groups were formed, and each group consisted of eight mice. However, according to the ethics committee’s advice, the number of mice was reduced to six in each group. The mice were grouped without randomization due to having the same characteristics. The three groups included the pcDNA3.1-*lipX* group, the pcDNA3.1 immunization group, and the without immunization group. All test animals underwent a process of acclimatization for two weeks in a cage regulated for temperature and air circulation. The maintenance of the cage was carried out twice a week, and food and drink were provided ad libitum. Retro-orbital blood collection, spleen isolation, and cage cleaning were carried out sequentially and carried out one by one between groups to avoid mixing mice and causing data collection errors from each group.

### 2.5. Procedure for Mouse Immunization and Blood Collection

Six experimental mice in each group were immunized with 100 μg of pcDNA3.1-*lipX* or 100 μg of pcDNA3.1 in 100 μL of saline per injection, respectively, and one group was not immunized. Immunization was carried out intramuscularly using a 27G syringe three times within two weeks, and the mice were terminated one week after the last immunization (Appendix C). The protein concentration and duration of immunization until termination were based on a previous study [33,34]. The mice were anesthetized with ketamine-xylazine collection before every retro-orbital blood collection and immunization. Sera from blood collected before the third immunization (booster II) were analyzed for their humoral immune response against LipX-His protein through Western blot, and splenocytes from terminated mice were analyzed for their cellular immune response against LipX-His through enzyme-linked immunosorbent assay (ELISA).

### 2.6. Examination of Humoral Immune Response Using Western Blotting

LipX-His protein was obtained from a DNA recombinant of pET28a-*lipX* in the *E. coli* BL21 strain (unpublished data). This protein was expressed by the induction of 1 mM of isopropyl-β-d-thiogalactopyranoside (IPTG) (Thermo Fisher Scientific, Waltham, MA, USA) and isolated using an Ni-NTA column. LipX-His protein was run through sodium dodecyl sulfate-polyacrylamide gel electrophoresis (SDS-PAGE) and transferred into a PVDF membrane, which was followed by 2 h of incubation with 5% skim milk at room temperature (RT) in a 50 rpm shaker. The membrane was then incubated with 1:1000 diluted pooled mice sera, which was followed by incubation with 1:5000 diluted goat anti-mouse IgG-labeled peroxidase (Invitrogen) for 1 h each at RT in a 50 rpm shaker. As a positive control, an antibody anti-His protein conjugate with peroxidase (Thermo Fisher Scientific, Waltham, MA, USA) was also added to other similar membranes. TBS-tween 0.1% was added to each incubation step for 5 min, and this step was repeated three times. The protein band was visualized by the addition of a TMB substrate (KPL).

### 2.7. Cellular Immune Mediator Response Assay

One week after the last immunization, the mice were terminated by cervical dislocation; the spleens were then isolated aseptically. Spleens were homogenized in RPMI medium containing 1% antibiotics (Gibco) and 10% FBS (Gibco) to obtain 2 × 10^5^ cells/mL of splenocyte suspension. Splenocytes were then plated in 96-well culture plates with an RPMI-1640 media (Gibco) and divided into three groups: induced by LipX-His protein 507.28 ng/µL, induced by PHA, and noninduced. Splenocytes were incubated at 37 °C with 5% CO_2_ for 72 h. After incubation, the supernatant was collected for IFN-γ, IL-12, IL-4, and IL-10 analysis by ELISA (Thermo Fisher Scientific, Waltham, MA, USA).

### 2.8. Statistical Analysis

Data from the measurement of cytokines IL-4, IL-10, IL-12, and IFN-γ were statistically analyzed using the two-way analysis of variance (ANOVA) test for normally distributed data and the Kruskal–Wallis test for nonnormally distributed data. Significant differences were determined at *p* < 0.05.

### 2.9. Limitations of the Study

This study was conducted only on the BALB/C mice strain. Therefore, the resulting data may be different when applied to other test animals or humans.

## 3. Results

### 3.1. LipX Isolation and Observation of Humoral Immune Response by Western Blotting

Before observing LipX expression in the group of immunized mice, we isolated the LipX protein for use in the analysis of the humoral response. The isolation results using a Ni-NTA column showed that we successfully isolated the protein with the appropriate size for the length. The protein bands produced followed the expected protein length of about 13.34 KDa, which was similar to LipX-His protein’s length (Figure 1).

To determine whether there was an adaptive humoral response in the serum of mice injected with recombinant DNA pCDNA3.1-lipX as an indicator of LipX expression in vivo, we used Western blotting of LipX-His against immunized sera. Observation of the humoral response in the serum of mice in this step showed anti-LipX antibody binding marked by a protein band measuring about 13.4 KDa, which was equivalent to the size of the LipX-His protein detected by the anti-His monoclonal antibody. The presence of this band indicated that pCDNA3.1-lipX was successful in expressing LipX in vivo (Figure 2). These results also prove that LipX can stimulate an adaptive humoral immune response in test animals.

### 3.2. Analysis of the Cellular Immune Mediator Response Assay

Identification of the cellular immune mediator to LipX was carried out using ELISA. The analysis was undertaken by measuring the concentration of selected cytokines, specifically IL-4, IL-10, IL-12, and IFN-γ from the splenocyte cultures of mice immunized with pcDNA3.1-*lipX*, pcDNA3.1, and without immunization (Figure 2 and Appendix A). Based on the results of statistical analysis, it was shown that there was a significant difference between the splenocytes of mice in the pcDNA3.1-lipX immunization group induced by LipX-His and the splenocytes induced by PHA and without any induction.

The LipX-His protein induced the splenocyte culture from pcDNA3.1-*lipX* immunization group to have the highest levels of IFN-γ cytokines (mean + SD: 441.222 + 54,548) compared to the splenocyte culture induced by PHA (mean + SD: 307.222 + 51,768) and without any induction (mean + SD: 307.222 + 51,768) in the immunization group (mean + SD: 258,555 + 16.431). Based on statistical analysis, a significant difference in the IFNγ titer was obtained between the splenocytes from the pcDNA3.1-*lipX* immunization group which was induced by LipX-His and the splenocytes induced by PHA and without any induction, at a *p* value of 0.000.

In the measurement of IL-4 titers in all groups of mice, it was found that the PCDNA3.1-lipX immunization group induced by LipX-His had the highest levels of the IL-4 cytokines titer (mean + SD: 59.881 + 8.32) compared to the PHA-induced splenocyte culture supernatant (mean + SD: 39.441 + 7.89) and splenocyte culture supernatant without induction (mean + SD: 32.017 + 2.50). Statistical data showed a significant difference between the pCDNA3.1-lipX immunization group, the pCDNA3.1 immunization group, and the nonimmunized group (*p* = 0.000).

A significant difference was also found in the production of IL-12 cytokines in the PCDNA3.1-*lipX* immunization group which was induced by LipX-His, where the levels of the IL-12 cytokines titer (mean + SD: 221.142 + 46.235) were higher than those of the splenocyte culture induced by PHA (mean + SD: 54,857 + 13.982) and without any induction (mean + SD: 52,285 + 11,926). Statistical analysis showed a *p* value of 0.000 for this cytokine titer comparison.

No statistically significant difference was found between groups of mice in the IL-10 titer measurement; the statistical analysis showed a *p* value of 0.232 (Figure 3).

## 4. Discussion

As the only available vaccine against *M. tuberculosis* infection, BCG has several drawbacks, from varied efficacy in healthy adults to safety concerns in immunosuppressed individuals. Thus, many researchers have sought to find an alternative tuberculosis vaccine to be used together with or to replace the BCG vaccine. LipX is a member of the PE protein family, which is rich in proline amino acids and responsible for lipid remodeling in the *M. tuberculosis* cell wall in terms of hindering immune evasion [25]. Furthermore, in silico analysis of this protein has proven that LipX tends to have a recognition domain for B cells and major histocompatibility complex (MHC) class I [26]. In this study, we evaluated LipX immunogenicity in the form of a DNA vaccine.

In detecting anti-LipX IgG in the serum of mice, an IgG-specific response against LipX was detected by HRP-labeled horse anti-mouse IgG. The interaction between the LipX-His protein, mouse IgG, and HRP-labeled anti-mouse IgG on the PVDF membrane was indicated by the presence of a specific band that appeared after the TMB substrate was added. This band’s appearance indicated that our recombinant pcDNA3.1-*lipX* plasmid could express recombinant LipX protein in vivo. Although the presence of the LipX antigen in mice was not directly analyzed, a specific IgG response against LipX was successfully proven using Western blot, indicating the antigenicity of the expressed LipX protein. This specific humoral immune response was formed due to the presence of antigen presentation by APC cells or infected somatic cells [13].

In terms of the DNA vaccine, our pcDNA3.1-*lipX* would be taken up by muscle cells and neighboring dendritic cells before immunization. In theory, LipX will be expressed and presented to the cell’s surface through MHC or secreted outside the cells. Alongside the in silico result, we assumed that our LipX protein would be presented by MHC class II to further activate the T cell response and initiate cellular immunity [27]. However, secreted LipX can also be recognized by antibodies or B cells. Generally, in *M. tuberculosis* infection, the activation of B cells through Th1 or Th2 cells in the presence of IL-4 signals is received by B cells. The formation of plasma cells leads to secretion of specific IgG antibodies against antigens [13,33]. Moreover, even though we did not perform an analysis of activated T cells in the serum of immunized mice, the Western blot analysis results might indicate T cell activation. Based on these data, we conclude that the duration from immunization until termination of the mice was sufficient to detect a cellular immunity response.

The cellular immune response is the most crucial line of defense against *M. tuberculosis* [35,36]. This response is characterized by the host’s ability to act to develop a subset of Th1 and Th2 cells, thereby leading to the continuous activity of macrophages and dendritic cells. In evaluating cellular immunity against the DNA vaccine seed, pcDNA3.1-*lipX*, ELISA was carried out to determine the cytokine levels significant in adaptive immunity during infection. The activation and performance of macrophages are essential in fighting infection from *M. tuberculosis*, an intracellular microbe [27].

There are two types of cellular immunity reactions that arise when dealing with microbes. Specifically, these include CD4+ T helper cells that secrete mediators that recruit cytokines, thus activating other leukocytes in the phagocytes of pathogenic microbes. For example, they are activated to defend against intracellular microbes, such as *M. tuberculosis*. CD8+ cytotoxic T lymphocyte (CTL) cells can also be activated. CTL cells kill infected cells or contain pathogenic proteins, such as viruses, that infect cells in the cytosol [27]. In the case of *M. tuberculosis* infection, the dominant type of cellular immune reaction is the type that leads to macrophage activity [37,38]. The cytokines that mediate macrophage activation are the IFN-γ and TNF-α cytokines secreted by the Th1 cell subset [39].

The cellular immunity results obtained through ELISA in this study showed a significant increase in cytokine levels in the splenocyte culture samples of mice immunized with pcDNA3.1-*lipX* and induced by recombinant protein LipX-His compared to those that were induced by PHA or not induced. The cytokines that showed increasing levels were IL-4, IL-12, and IFN-γ. This is consistent with the host’s defense mechanism against *M. tuberculosis* infection [27,39]. These three cytokines play a synergistic role in fighting infection. Meanwhile, the titers of IL-10 did not show a significant increase in the splenocyte cultures of mice immunized with pcDNA3.1-*lipX* versus those induced with LipX-His protein and not induced. IL-10 cytokines are known to have antagonistic functions with the three cytokines described above [27,40]; IL-10 cytokines inhibit the production of cytokines and chemokines and reduce the expression of stimulators and MHC-II molecules. The main targets of IL-10 cytokines are macrophage and dendritic cells; in other words, IL-10 directs the failure of the host defense process against *M. tuberculosis* infection [27,41]. Although IL-10 will naturally indeed be produced in instances of prolonged inflammation, it acts as a form of physiological balance for the body in promoting anti-inflammation [42].

Cellular immune responses occur in several stages, starting with antigen-stimulated T cells presented by APC cells through MHC class I or class II [24]. In peripheral lymph nodes, the introduction of antigens through MHC-II activates CD4+ T cells to secrete IL-12, which causes the development of subsets of Th1 that will produce IL-2, IFN-γ, and TNF-α, which will send signals to macrophages and dendritic cells to be more active in the phagocytosis of pathogens. Additionally, CD4+ T cells can secrete IL-4 cytokines, which can activate the development of the Th2 subset, thus producing IL-4, which targets B cell activation to activate or mediate adaptive humoral immunity [27,28].

As mentioned in the introduction, LipX plays a role in hindering macrophages. Research conducted on *M. smegmatis* showed that several genes in the pe family, including lipX, play a role in macrophage intracellular survival [25,43]. Overexpression of LipX in experimental *M. smegmatis* (Msmeg-PE11) causes changes in colony morphology and lipid composition of cell walls, and envelopes of this strain have a higher number of glycolipids and polar lipids; the bacteria become more difficult to kill by macrophages thereby increasing survival [25]. Mice immunized with *M. smegmatis* expressing LipX also had high bacterial counts, organ damage, and death [25]. LipX expression also decreased interleukin 6 levels [25,44]. These data indicate that LipX protein plays a role as a virulence factor. We assume that, based on the status of the analyzed cytokines and the results of humoral analysis, the pcDNA3.1-*lipX* DNA vaccine seed has the potential to be developed as a TB vaccine candidate. However, vaccination may be a mediation for bacteria in avoiding macrophages and increasing the number of bacteria, especially in hosts that already contain *M. tuberculosis*, for example, in latent tuberculosis. This assumption may indicate the pcDNA3.1-*lipX* vaccine only can be administered as the first vaccination in newborns and not as a booster for the BCG vaccine or for the vaccination to adults who might have been exposed to *M. tuberculosis* previously. In addition, due to LipX only being conserved in the *M. tuberculosis* complex and some clinical isolates, it might be that pcDNA3.1-*lipX* is only effective against infection of those species.

In addition, since we have not carried out the challenge study, some information, such as the effectiveness of the recombinant DNA of pcDNA3.1-*lipX* in inhibiting the growth of *M. tuberculosis* in the host, has not been confirmed. This study is limited to information on the ability of our constructed pcDNA3.1-*lipX* to stimulate humoral and cellular immune responses in the splenocyte culture of BALB/c mice. We also assume different strains of mice in the experiment may also reflect different humoral and cellular responses.

## 5. Conclusions

Our seed vaccine construction, pcDNA3.1-*lipX,* has potential as a candidate tuberculosis vaccine. This argument is based on its ability to induce cellular immune reactions, which are the primary protection against *M. tuberculosis* infection. However, further studies, such as challenge testing and the effect of LipX on *M. tuberculosis* survival in macrophage, are needed to strengthen information on this seed vaccine’s power to protect against disease in the host.

## Figures and Tables

**Figure 1 genes-13-01954-f001:**
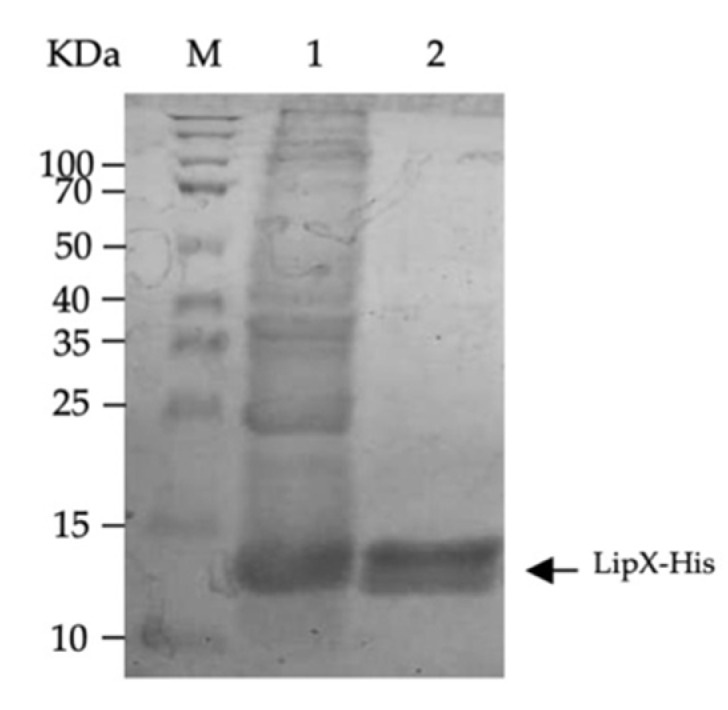
LipX-His protein isolation. Lane 1, cell lysates from the 1 mM IPTG induction culture of *E. coli* BL21 containing the pET28-*lipX* plasmid; Lane 2, Protein isolated using the Ni-NTA affinity column from the culture of *E. coli* BL21 containing the pET28-*lipX* plasmid inducted by 1 mM of IPTG. KDa—kilo Dalton; M—marker.

**Figure 2 genes-13-01954-f002:**
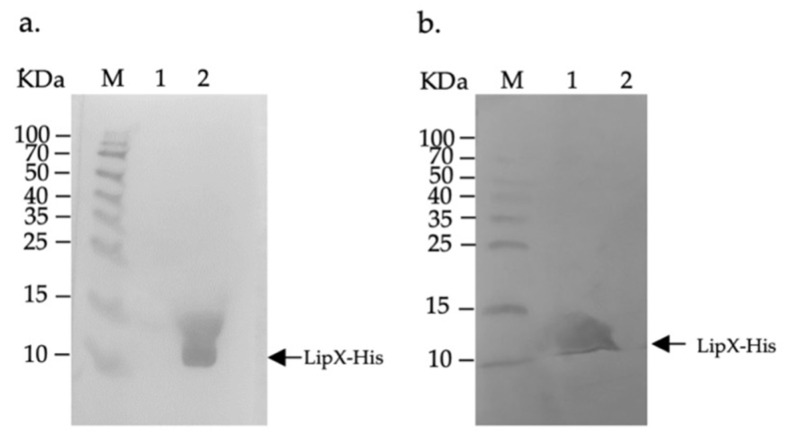
Analysis of the humoral immune response using Western blotting. (**a**) Confirmation of LipX-His protein expression using the HRP-labeled anti-His antibody detector: Lane 1, cell lysate from the culture of *E. coli* BL21 containing the pET28a-*lipX* plasmid without IPTG induction; Lane 2, cell lysates from the IPTG induction culture of *E. coli* BL21 containing the pET28a-*lipX* plasmid; (**b**) Proving the humoral immune response using serum pcDNA3.1-*lipX*-immunized mice to determine the expression of the LipX-His protein: Lane 1, cell lysates from the IPTG induction culture; Lane 2, cell lysates from the culture without IPTG induction. KDa—kilo Dalton; M—marker.

**Figure 3 genes-13-01954-f003:**
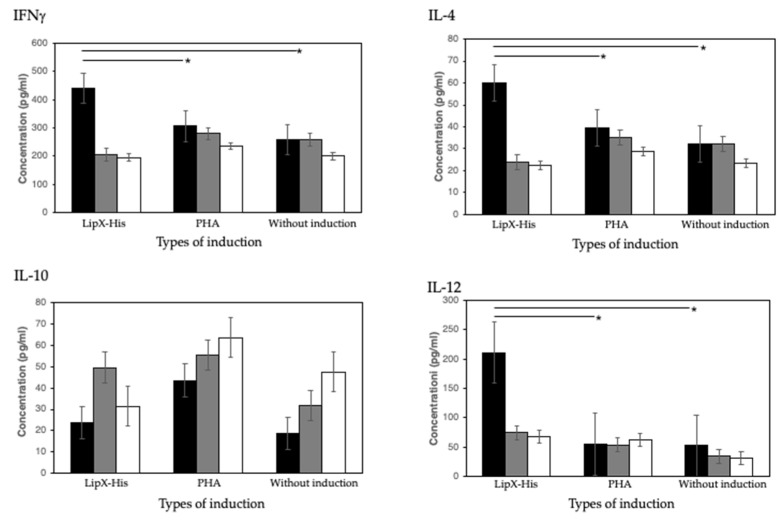
Concentration of IFN-γ, IL-4, IL-10, and IL-12 between immunization groups following the stimulation of splenocytes with LipX-His or PHA, as well as without induction. Black, splenocytes from mice immunized with pcDNA3.1-*lipX.* Gray, splenocytes from mice immunized with pcDNA3.1. White, splenocytes from mice without immunization. Asterix label (*) indicates a statistically significant difference.

## Data Availability

Not applicable.

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
