# Peer review of "Responses of Humoral and Cellular Immune Mediators in BALB/c Mice to LipX (PE11) as Seed Tuberculosis Vaccine Candidates"

_genes, 2022, doi:10.3390/genes13111954_

Round 1

Reviewer 1 Report

Authors conducted very basic study to evaluate the efficacy of LipX protein as a vaccine candidate against M. tuberculosis. They vaccinated BALB/c mice and analyzed humoral response as well the cellular immune response by estimating the levels of IFN- γ, IL-4, IL-10, and IL-12 in the spleenocytes culture supernatants.  Authors have to address the following concerns.

1.     Mice were immunized with 100 μg of pcDNA3.1-lipX or 100 μg pcDNA3.1 per injection intramuscular. How did you select the dose of vaccine as well as the mode of injection?

2.     Mice were vaccinated three times within two weeks. Authors should specify the rationale of the study design. Please specify minimum time interval time between booster doses. 

3.     Did the authors estimate the systemic (plasma/serum) level of Th1 and Th2 cytokines response before and after vaccination, in addition to the spleenocyte response?

4.     Authors should analyze the data within same stimulation group (example: data analysis between the mice immunized with pcDNA3.1-lipX vs mice immunized with 186 pcDNA3.1 or without immunization. I suggest you incorporate statistical significance in figure 2.

5.     Authors have shown increased IFN-g and IL-12 (is it IL-12a?) by spleenoctyes. Did you study whether Lip-X induced cytokine production affected intracellular survival of M. tuberculosis in macrophages?

6.     It is better to revise the sentence “As an intracellular pathogen, M. tuberculosis infection control depends on the development of an adaptive immune response in the humoral and cellular stages at the site of infection in the host” (line #56-8).

7.     Authors mentioned that Lip-X is considered one of M. tuberculosis’s factors in hindering host immune evasion, especially by macrophage cells. Please specify whether Lip-X contribute to immune evasion or hinder from immune evasion? Not clear. Athor should explain the mechanism how Lip-X modulate host immune response, which underscore the importance this study.

8.     Figure-1A – line is not marked

Reviewer 2 Report

1. please go for a proof reading first, the English is difficult to understand

2. the design lack essential support (challenge experiment) for the final conclusion, 

3. lots of key results are missing, whereas methods part mentioned

4. figures are not well presented, ligends did not in detailed

5. presentation is not consistent, such as TB is disease or bacteria?

6. why using spleen from different groups and induced by lipX? it is not clearly explained. why not using MTB to induce?

7. need controls, such as lipX as vaccine, BCG as control

etc. 

I have made some comments in the pdf file, please consider.

Reviewer 3 Report

Major comment

Vaccine development has expectations for population treatment globally and this is particularly important for control of intractable diseases, including the disease caused by Mycobacterium tuberculosis. The research reported in this manuscript detailed novel antigen selection associated with producing a vaccine for this pathogen; furthermore, immunization using the developed plasmid appears to be effective. However, I have some questions about whether the proposed antigen candidate works effectively as a vaccine, and it is necessary for the authors to consider the following points.

The possibility of establishing anti-tuberculosis immunity is only envisaged based on cytokine expression and protein expression, although I cannot say if this was successful. As highlighted in the Discussion, the production of IgG against antigen is important in the case of humoral immunity while CD8-positive memory T lymphocytes are also important for cellular immunity. Cellular immunity seems to be established through cytokine secretion in this study, but there is a need to discuss whether these memory mechanisms work. It is also important whether this memory is maintained long term.

Regarding the humoral immunity response to lipX-HIS, it is necessary to consider how lipX-HIS is processed in vivo.

Minor comment

1.    Although lipX is known to be highly toxic, there is no cytokine movement in non-immunized mice; therefore, it is necessary to verify whether the lipX used in the study is effective.

2.    If immunity to tuberculosis is established by the author’s antigen, it seems that Th1 and Th2 reactions are occurring at the same time because both IL-12 and IL-4 were generated from the same sample.

Is this consistent with biological reactions?

3.    Has the safety of the vector used been established in humans? If there are any associated side effects, please mention these as well.

4.    How many other strains are covered by antigen immunization from the tuberculosis Beijing strain?